# Soil Organic Matter of Tidal Marsh Permafrost-Affected Soils of Kolyma Lowland

Vyacheslav Polyakov [1,*], Alexey Lupachev [2], Stanislav Gubin [2] and Evgeny Abakumov [1]

1    Department of Applied Ecology, Faculty of Biology, St. Petersburg State University, 16th Liniya V.O., 29, 199178 St. Petersburg, Russia

2    Institute of Physico-Chemical and Biological Problems in Soil Science, Russian Academy of Sciences, Institutskaya St., 2, 142290 Pushchino, Russia

*    Correspondence: slavon6985@gmail.com; Tel.: +7-953-1724-997

**Abstract:** Soils of the Arctic sea coasts are one of the least studied due to the complex logistical accessibility of the region, as well as the severe climatic conditions. The genesis of these soils is determined by several factors of soil formation simultaneously—cryogenesis, the influence of river alluvial processes, as well as the tidal influence of the sea. The paper presents data on the morphological structure of soils formed on the seacoast of the East-Siberian Sea (Kolyma Lowland, North Yakutia). Under the influence of cryogenesis and sea water tidal input, marsh soils are formed, with a relatively high level of salinity and the development of gleyization. Autochthonous and allochthonous soil organic matter play a leading role in marsh soil formation here, including the possible accumulation and biochemical transformation of incoming pollutants (e.g., hydrocarbons). The main objective of the study was to evaluate the soil organic matter genesis and alteration under the influence of tidal processes in coastal permafrost-affected soils as well as to obtain the previously unknown characteristics of the structural and elemental composition of different fractions of organic matter. The elemental composition and $^{13}$C NMR spectroscopy of humic acids were analyzed. It was revealed that humic acids extracted from the studied marsh soils accumulate up to 50% C and 4% N. Active processes of dehydrogenation are noted in HAs molecules, which indicates a relatively low degree of aliphatic structure development. According to $^{13}$C NMR spectroscopy, it was revealed that up to 45% of aromatic structural fragments accumulate in marsh soils, indicating a relatively high degree of organic matter stabilization and resistance to biodegradation.

**Keywords:** $^{13}$C NMR spectroscopy; humic acids; maritime soils





## 1. Introduction

Occupying vast areas, soils of the seacoasts play an important environmental role. Seashores of the accumulative type are subdivided into marsh shores subjected to flooding by tidal or surging seawater with marsh soils and nonflooded areas influenced by the sea owing to impulverization of salts, groundwater recharge, the specificity of soil-forming marine sediments, etc. The soils developing in non-flooded coastal areas are classified as maritime soils, and all the soils formed under the direct or indirect influence of the sea were previously proposed by Shlyakhov to be combined into a general group of Thalassosols [1]. Soils of the Arctic Ocean coasts remain poorly studied and underrepresented in the majority of the general and specialized soil maps [2] and soil classification systems [3]. The large length of the Arctic coastline, the diversity of coastal geomorphic features and plant cover, sediments, the character of tidal and surge phenomena and the salinity of the seawater determine the formation of a wide range of soils [4]. Soil morphology and properties on the Arctic coasts are complicated due to the severity of the climate, cryogenic processes, specific microbiota, flora and fauna and other environmental factors [5–9].

Within the marshes of the East-Siberian Sea coastal part of the Kolyma Lowland (North Yakutia, Russia), tidal flat seashores (watts) stand out as the areas regularly submerged

under seawater, even during low tides or seawater surges. They are often vegetation-free or covered with individual spots of halophytes. Saline or herbaceous meadows with the participation of halophytes grow in areas that are situated farther from the sea and are less often affected by seawater at the territory of marsh areas. Marsh territories of the Arctic accumulative shores can extend tens of kilometers from the seacoast. Among the marsh and tidal flat soils, there are soils partially or nearly completely consisting of organic material: peat, detritus, a humified mass of sea algae and even considerable masses of wood debris [10].

Thus, autochthonous and allochthonous soil organic matter (SOM) plays a leading role in marsh soil formation, including the possible accumulation and biochemical transformation of incoming pollutants (e.g., hydrocarbons). In the conditions of increasing economic activity in the Arctic seas, the threat of the pollution of fragile, permafrost-affected marsh soils adjacent to the coasts is sharply increasing [11].

Taking into account the wide geographical distribution of tidal marsh permafrost-affected soils, a considerable amount of the SOM migrate and accumulate within the soil profiles every year due to the processes of cryogenic mass exchange, the tidal input of mineral material, plant detritus and dissolved substances [12]. Rapid changes in the high-latitudinal environments make the issues of the intraprofile transformation and stabilization of organic substances more and more challenging. In this context, the investigation of the SOM stabilization rate and humification degree became urgent for the qualitative assessment of SOM storages in permafrost-affected soils [13]. Not only is quantity (stocks, percentage, etc.) an important index in the evaluation of the SOM stabilization rate, but the qualitative indexes—the degree of aromatization, the humification rate and the level of biochemical activity—are as well [14].

One of the methods for analyzing the degree of SOM stabilization is the analysis of the molecular composition of humic acids (HAs) isolated from the soil [15]. Humic substances are understood as a group of dark-colored organic substances that have similar structural features [16]. Their composition and properties are determined by local conditions of formation (climate, precursors of humification, composition and activity of soil microbiota). Humic substances consist of humic and fulvic acids, as well as a non-hydrolyzable residue. The high activity of humic and fulvic acids is due to a large set of functional groups—carboxyl, phenolic, alcoholic, amide, quinone and amine—capable of forming electrovalent, covalent and intracomplex compounds. Polydispersity and polyfunctionality provide a high buffering capacity of these molecular systems with a relation to the acid-base, redox and many others. HAs play a significant conservation role, providing stable features to soils and thus participating in the deposition of SOM [17]. In Arctic environments, an important role of humic substances is climate regulation. This is associated with the formation of biotic- and abiotic-resistant compounds in HAs.

Soils of the coasts of the Arctic seas (marshes and watts) are still extremely poorly studied. The nature and properties of their SOM are poorly known, which makes it difficult to assess the possible pollution of these soils and to develop approaches and methods for their remediation. Thus, the main objective of this study was to evaluate the SOM genesis and alteration under the influence of tidal processes in coastal permafrost-affected marsh soils as well as to obtain the previously unknown characteristics of the structural and elemental composition of different fractions of SOM.

## 2. Materials and Methods

### 2.1. Study Area

The soils of the accumulative coasts of the East Siberian Sea in the Arctic tundra zone in the north of the Kolyma Lowland, in the estuary part of the Bol'shaya Chukochya River (the eastern sector of the Russian Arctic), were studied (Figure 1). The morphology and properties of these soils are greatly complicated by the severity of the climate, permafrost conditions and cryogenic processes. Most of the coasts of the East Arctic seas are represented by the shores of the accumulative type with small slopes, which leads to the

penetration of tidal and surge seawater into the land for tens of kilometers, so seawater feeds numerous small lakes, low lake terraces and permafrost polygons. We also analyzed the soils of tidal flats, marshes and adjacent maritime areas that are not submerged under the seawater nowadays but are influenced by the wind input of salts and mineral material from the seaside.

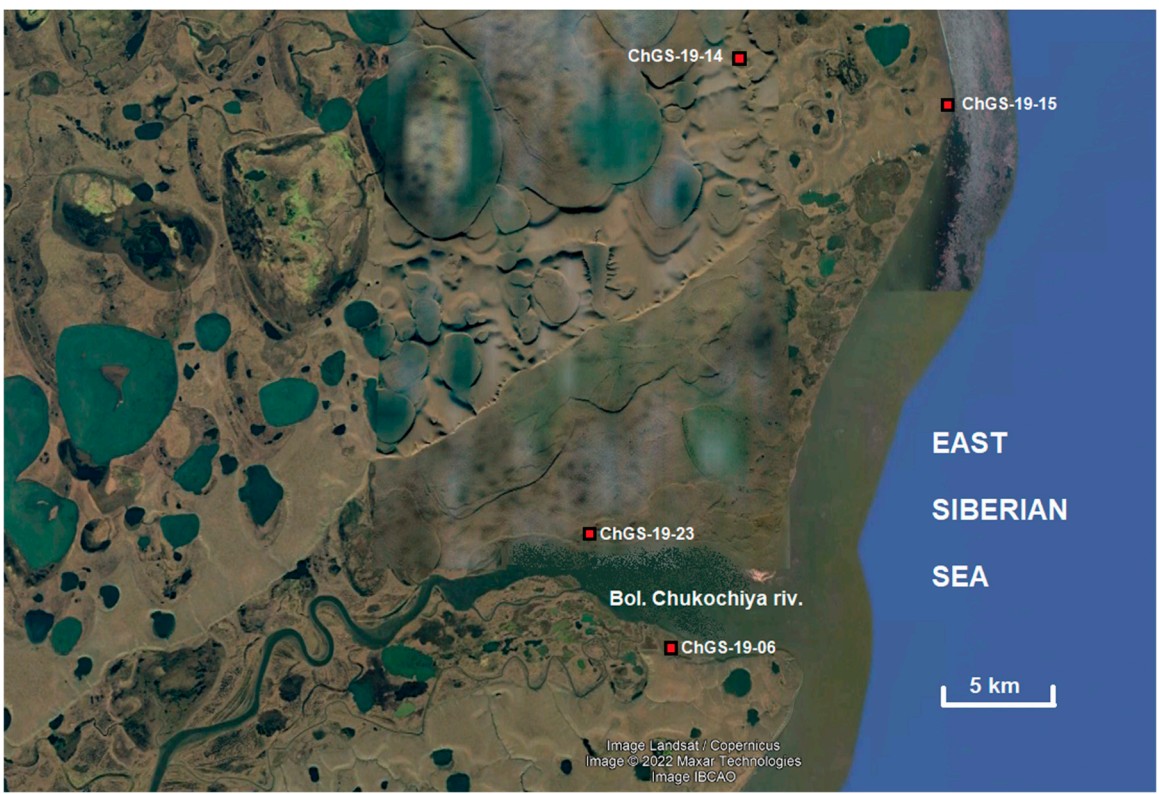

**Figure 1.** Watt (ChGS-19-06), marsh (ChGS-19-23), maritime soils (ChGS-19-14) and allochtonous deposits sampling point (ChGS-19-15) of the studied area.

The coastal area in the studied region is characterized by the maritime arctic climate, [18] with long and cold winters (mean January temperatures of (−35)–(−37) °C), short summers (with mean July temperatures of 8–12 °C) and annual precipitation of about 150 mm, 30% of which falls in summer, with fogs and drizzling rain. The snow cover remains from October to June. Rivers and the sea become ice-free in the second half of June [19]. Often, during the summer, large masses of sea ice are carried by storms and wind surges to low shores, where they melt. In summer, in the coastal part of the sea, the water temperature is 4–7 °C. The water level rise during extreme surges can exceed 1.5 m, with frequent ordinary water rises of 20–50 cm. The thickness of the permafrost active layer in the coastal zone is 30–80 cm. Gradual soil thawing begins after the snowmelt season in the middle of June.

The territory under study mainly represents the valley of the Bol'shaya Chukochya River; it gradually expands towards the sea and reaches more than 10 km in the coastal part with tidal flats and tidal marshes (Figure 2). It is bound by high coasts composed of silty sediments of the ice complex (yedoma) containing relict organic matter and subjected to active degradation [20,21].

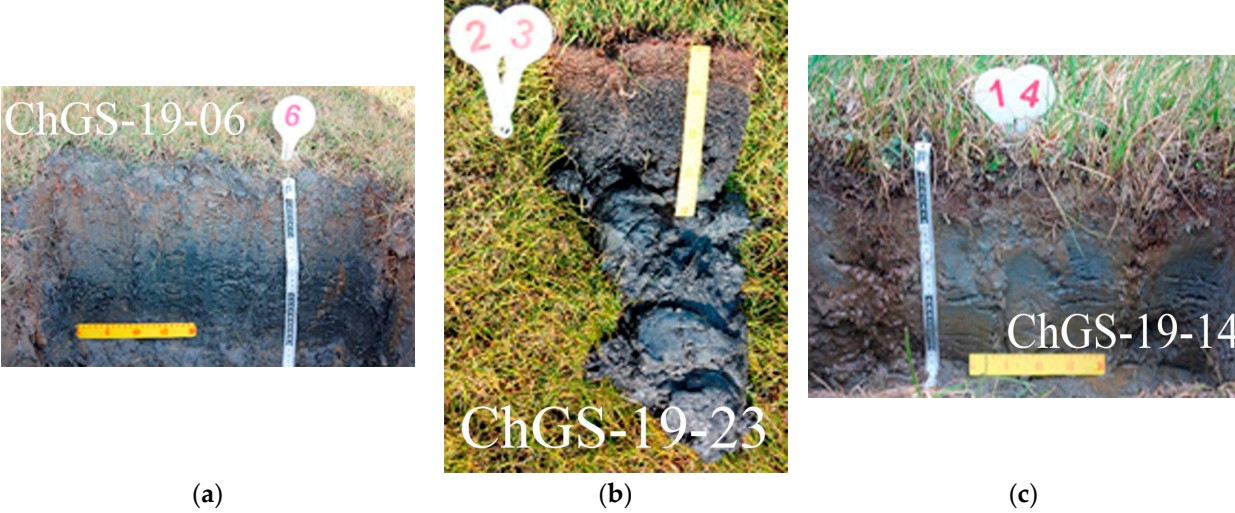

**Figure 2.** Watt (**a**, ChGS-19-06), marsh (**b**, ChGS-19-23) and maritime (**c**, ChGS-19-14) soil morphology.

On the low accumulative coasts of the Bolshaya Chukochia River bay, we studied permafrost-affected soils on watts with sparse vegetation cover—ChGS-19-06 (N 70°5′22.54″ E 159°49′37.66″)—and on the marsh territories covered with graminoids, sedges and halophytes—ChGS-19-23 (N 70°7′6.27″ E 159°49′49.12″). The width of the water area at the mouth of the river is about 2 km. Inland, the surface of the low accumulative coast of the bay has a very gentle (<1°) slope. Near the sea, its height marks are 0.2–0.5 m above sea level (a.s.l.); at a distance of 10 km, they are 1.2–1.5 m a.s.l. During high surges, seawater can penetrate inland for more than 10 km, flooding the adjacent vast flat territory of tidal marshes and the river floodplain. In the coastal part of the bay, the salt content in water in August is about 3.0‰, while the average value in the sea at a distance of 2 km from the coast is around 3.6‰. According to water extract data, the surface layers of bottom sediments are slightly saline (0.2–0.5‰). Fine sand predominates in their composition, while coarse silt predominates in the soils of tidal marshes. The soils and bottom sediments of the coastal zone are characterized by a low content of carbonates (<0.9%). The studied soils are characterized by the chloride-sulfate magnesium-sodium type of salinity. Additionally, we studied maritime permafrost-affected soils in small river channels (rivulets), through which seawater penetrates deep inland into coastal lakes during high surges—ChGS-19-14 (N 70°16′22.14″ E 159°51′21.36″). Higher-located tidal marshes were studied there. The water salinity in the rivulets is 0.3–0.5‰, Maritime soils were studied on the sea coast and along the channels of the rivulets at a distance of 0.2 to 0.5 km from the marsh zone. The salinity of sea water 200 m away from the estuaries of rivulets is 3.7–4.2‰. In the tidal shoreline, the material of the allochtonous marine-derived plant detritus—ChGS-19-15 D—and the silt deposit—ChGS-19-15 H (N 70°15′15.47″ E 160°5′42.35″)—was also sampled and analyzed. Accumulative shores of the Kolyma Lowland with shallow slopes that are subjected to the tidal processes are mainly divided into two major groups in a pedological sense: watts—the lowermost shores that are regularly affected by the tides and the input of the allochtonous mineral and organic matter, with sparse, mainly halophyte vegetation and poorly or unexpressed microrelief—and marshes—slightly more (0.5–1 m) elevated shores that are affected by the marine processes only during the high tides, with well-developed vegetation cover and a pronounced permafrost-affected pattern of polygonal microrelief. The studied soil pits were established in order to represent the morphology and physico-chemical properties of the soils in these two main maritime environments.

Field studies were carried out at the end of August, during the period of the maximum thawing of soils at a low sea level, which exposed the territory of watts and marches. The cryogenic microtopography, vegetation cover, active layer thickness and morphology of soil profiles were described along the transects and in key coastal areas. Soil temperature

measurements were performed with a thermal probe with a step of 5 cm down the soil profiles. We also measured temperature, pH and electrical conductivity in the water of marsh lakes, rivers and flooded polygons, as well as in the water appearing in soil pits. Samples of sediments, soils and water bodies were taken for further analytical studies. For the purpose of standardization, the drying of soil and sediment samples saturated with water or having a high ice content was carried out after the gravitational water was drained from them.

The authors used both Russian [3] and European [22] soil classification and diagnostic systems to name soils and soil horizons, indicating the participation of tidal synlithogenic processes, such as the inflow and accumulation of mineral and organic suspensions and salt solutions, that are superimposed on zonal soil formation processes.

### 2.2. Methods

Analytical studies of soils and sediments were carried out at the Center for Collective Use of the Institute of Physico-Chemical and Biological Problems of Soil Science of the Russian Academy of Sciences according to the methods generally accepted for soil studies [23]. Data on the particle size distribution of soil samples were obtained following the pipette sedimentation method using pyrophosphate dispergation [24].

### 2.2.1. Chemical Analysis

HAs were isolated according to the international protocol IHSS [25,26]. HAs are heterogeneous systems of high- and low-molecular-weight compounds formed by the decomposition of plant and animal residues in terrestrial and aquatic ecosystems. The elemental composition of humic acids (C, H, N, O) was determined by an elemental analyzer (Euro EA3028-HT Analyzer, Pravia, Italy). The high variability in the elemental composition of HAs among soils is explained by the bioclimatic features of the region, the precursors of humification and the degree of hydromorphism of the territory [27]. According to the elemental composition of HAs, we can determine the trends of humification occurring in the molecules. Information on the elemental composition of organic matter provides important information about the general principles of the molecular structure and their properties [16]. The van Krevelen diagram [28] is used for the graphical analysis of the elemental composition; H/C-O/C ratios are used to determine the direction of the transformation processes of various organic compounds in natural conditions. Thus, it is possible to estimate the direction of oxidation/reduction and hydrogenation/dehydrogenation processes in HA molecules. In the soils of the Arctic zone, the carbon content of HAs is significantly lower than that in the soils of more southern bioclimatic regions. This peculiarity is explained by the influence of the acidic pH reaction and the excessive moisture of the territory. There is also a lower content of nitrogen, which is associated with its low content in the composition of the precursors of humification (moss, lichens) and the increased content of hydrogen. Data on the elemental composition were calculated for a completely dry and ash-free sample. The oxygen content was determined by the difference in the ratios. Gravimetric concentrations are given for C, H, O and N content. C/N, H/C, O/C, H/C mod. and W were calculated from the mole fractions of C, H, O and N gravimetric content. H/C mod. is the number of substituted hydrogen atoms in HAs; H/C mod. = H/C + 2(O/C) × 0.67; the W index is the degree of oxidation in molecules. The H/C and W indices were calculated from [16]. SD ± 0.05 was employed for the contents of C, H and N. Information on the mass of extractable HAs is presented in Table 1.

**Table 1.** The basic information about HAs extraction.

| Soil ID | Soil Pit Index | Soil Horizon | Depth, cm | Mass of HAs, g | Extraction Yields of HAs, % |
|---|---|---|---|---|---|
| # 22 | | TE [1] | 0–9 | 0.17 | 5.78 |
| # 24 | | G [2] | 9–15 | 0.13 | 0.17 |
| # 25 | ChGS-19-06 | SS [3] | 15–25 | 0.12 | 17.58 |
| # 27 | | C [4] | 55–75 | 0.01 | 2.49 |
| # 45 | | TE | 4–8 | 1.01 | 11.60 |
| # 46 | ChGS-19-14 | G | 8–32 | 0.41 | 11.56 |
| # 47 | | D (detritus) | 0–5 | 0.37 | 1.19 |
| # 48 | ChGS-19-15 | H (silt deposit) | 0–5 | 0.03 | 1.24 |
| # 67 | | TE | 0–5 | 0.56 | 0.06 |
| # 68 | ChGS-19-23 | G | 5–20 | 0.62 | 0.92 |
| # 70 | | SS | 30–45 | 0.17 | 0.76 |

[1] Moderately decomposed organic material; [2] Gleyic horizon; [3] Pedogenetic accumulation of salts; [4] Parent material.

## 2.2.2. Procedure of CP/MAS $^{13}$C-NMR Spectroscopy

HAs were isolated from organic and mineral horizons of soils according to the method recommended by the International Society for the Study of Humic Substances, with a modification by Vasilevich [14]. HAs were isolated from air-dry soil samples by twofold extraction with 0.1 mol/dm$^3$ NaOH, after which a saturated solution of Na$_2$SO$_4$ was added to the alkaline extract to coagulate colloidal particles. The solid-state spectra of HAs were determined by CP/MAS $^{13}$C-NMR spectroscopy on a Bruker Avance 500 (Billerica, MA, USA) NMR spectrometer (in a 3.2 mm ZrO$_2$ rotor). The magic angle rotation rate was 20 kHz, and the nutation frequency for cross-polarization was u1/2p 1/4 62.5 kHz. The repetition delay was 3 s.

Numerous molecular fragments were identified by CP/MAS $^{13}$C NMR spectroscopy: carboxyl (-COOR), carbonyl (-C = O), CH$_3$-, CH$_2$-, CH- aliphatic, -C-OR alcohols, esters and carbohydrates, phenolic (Ar-OH), quinone (Ar = O) and aromatic (Ar-) groups, which indicates the great complexity of the structure of HAs and the polyfunctional properties that cause their active participation in soil processes. Six chemical groups in HAs were identified by $^{13}$C-NMR spectroscopy: nonpolar alkyl (0–46 ppm), N-alkyl/methoxyl (46–60 ppm), O-alkyl and anomer (60–110 ppm), aromatic compounds (110–160 ppm), carboxyl, esters, amides (160–185 ppm) and quinone (185–200 ppm).

## 3. Results

### 3.1. Morphology of the Studied Soils

In the estuary, tidal areas occupy the part of the coast adjacent to the sea, extending 300–500 m inland. Their surface is elevated above sea level by 20–30 cm and is completely flooded by tidal and surge water. Smooth and often without vegetation, the surface of tidal flats is complicated by small depressions, formed under the action of sea ice during their movement. Many small puddles and lakes filled with low-salinity water are scattered on the surface of the coast. The surface of the shoals consists of silty loam with an absolute predominance of coarse and medium silt, a low clay content (7–10%) and a high proportion of plant detritus. The material contains 1.5 to 5.0% C; the loss during calcination is 10%. The thickness of the active layer in August is 40–55 cm. The soil pit ChGS-16-06 was laid on the surface with a sparse cover of halophylic grasses, and a permafrost layer was found at a depth of 65 cm. The soil profile had a layered structure. Water appears from a depth of 40 cm. The lower soil layers contain total dissolved solids (TDS 5.6%) with a predominance

of sodium chlorides. The soil is classified as a poorly developed, layered, gley solonchak frozen marsh soil.

Soil pit ChGS-19-23 was laid 600 m from the seashore, on a surface elevated 80 cm above sea level and covered with sedges (*Scheuchzeria* sp.). The thickness of the active layer was 50 cm, and the water table was located at 10 cm. The soil can be classified as a saline frozen peaty-gleyey bog soil. It is characterized by a sharp decrease in C down the soil profile; the leaching loss also increases in the mineral horizons, except for their upper part, enriched with organic matter extracted from the surface peat layer. The acidic reaction of the upper organic horizon changes to neutral in the deep horizons. The TDS content in the peat horizon is 6.4%, and soil salinity belongs to the chloride-sulfate type. Among exchangeable cations, magnesium and calcium prevail, and the content of exchangeable sodium is also high. The soil is characterized by a high content of available phosphorus and potassium.

The soil pit ChGS-19-14 was laid at a distance of 10 km from the seacoast, near the river banks, in the absence of polygons under sedge-forb vegetation. With the participation of mosses, peaty and raw-humus, permafrost-affected marsh soils are developed, which are very rarely affected by the tidal processes. The profiles of these soils are relatively more drained (do not contain free water), and the active layer thickness is about 30–40 cm. The organogenic TE horizon is up to 10 cm in thickness and consists of poorly composed brownish peat densely penetrated by fine roots of herbs with a tendency for the formation of a crumb structure as it is mixing with the loamy material in its lowermost part. The underlying mineral gleyic G horizons have a mottled color pattern with bluish and ocherous mottles; the content of oxidized ocherous mottles increases in the lower horizons. Soil salinization is absent here (TDS < 0.1%). Such soils are weakly saline, have a well-pronounced polygonal cryogenic microrelief of the surface and develop in the transition zone from marsh soils to the zonal ones (unaffected by modern marine processes).

In the mouth part of the estuaries and the tidal flat seashores that are raised above sea level only by 20–30 cm and are completely covered by sea waters during the tide, the smooth and often bare surface is complicated by small depressions, forming under the wave and sea current processes and the action of squeezed sea ice blocks during their spring movements. There are many small puddles and lakes scattered on the surface and filled with slightly saline water (4–8‰). These tidal flats are covered with small and thin (up to 10 cm) patches of washed-in peat and areas of small-sized, poorly decomposed plant detritus deposition (ChGS-19-15 D). The surfaces of tidal flats consist of silt loam with an absolute predominance of coarse and medium silt, a low content of clay (7–10%) and a high proportion of fine plant detritus (ChGS-19-15 H). The material contains 1.5 to 5.0% TOC; the loss on ignition reaches 10%.

In the modern soil classification of the soils of Russia, soils experiencing the impact of the sea—thalassosols, or soils of tidal flats, as well as marine soils experiencing the impact of sea water drops—are absent. The organization of the soils distributed in the territory of the wats and marshes on the accumulative shores of the seas of the Eastern sector of the Russian Arctic allows us to consider the soils formed here in three pedogenesis trunks—synlithogenic, primary and organogenic. At the level of order in the trunk of synlithogenic soils, bog (riverbed) soils can be distinguished; in the trunk of initial (primary) soil formation, the order of underdeveloped boggy soils can be distinguished. Finally, a new order of allochthonous organic soils can be distinguished in the trunk of organogenic soil formation.

### 3.2. Elemental Composition of HAs

The elemental composition of HAs is the most important indicator determining the hydrogenation, oxidation, reduction and dehydrogenation in molecule HAs [16,29]. The characteristic features of HAs formed in cold conditions—in particular, in soils affected by permafrost—are the relatively high H content and the reduced O content compared to the HAs of the boreal and subboreal soils.

The obtained data on the elemental composition, molar ratios and oxidation degree of HAs (W) are presented in Table 2.

**Table 2.** Elemental composition of the studied HAs of isolated soils.

| Soil ID | Soil Pit Index | Soil Horizon | Depth, cm | C | H | N | O | C/N | H/C | O/C | H/C mod. | w |
|---|---|---|---|---|---|---|---|---|---|---|---|---|
| # 22 | | TE | 0–9 | 36.40 | 5.47 | 3.63 | 54.50 | 11.00 | 1.79 | 1.12 | 3.30 | 0.45 |
| # 24 | ChGS-19-06 | G | 9–15 | 28.21 | 4.91 | 2.83 | 64.05 | 11.00 | 2.07 | 1.70 | 4.36 | 1.34 |
| # 25 | | SS | 15–25 | 38.75 | 5.52 | 3.65 | 52.08 | 12.36 | 1.70 | 1.01 | 3.05 | 0.32 |
| # 27 | | C | 55–75 | 30.57 | 4.45 | 4.76 | 60.22 | 7.49 | 1.74 | 1.48 | 3.72 | 1.22 |
| # 45 | ChGS-19-14 | TE | 4–8 | 50.01 | 5.95 | 3.73 | 40.31 | 15.61 | 1.42 | 0.60 | 2.23 | −0.21 |
| # 46 | | G | 8–32 | 14.45 | 4.20 | 1.40 | 79.95 | 12.05 | 3.46 | 4.15 | 9.02 | 4.84 |
| # 47 | ChGS-19-15 | D (detritus) | 0–5 | 52.78 | 5.74 | 3.09 | 38.39 | 19.92 | 1.30 | 0.55 | 2.03 | −0.21 |
| # 48 | | H (silt deposit) | 0–5 | 43.48 | 5.30 | 4.22 | 47.00 | 11.98 | 1.45 | 0.81 | 2.54 | 0.17 |
| # 67 | ChGS-19-23 | TE | 0–5 | 47.91 | 6.01 | 4.78 | 41.30 | 11.67 | 1.49 | 0.65 | 2.36 | −0.20 |
| # 68 | | G | 5–20 | 47.46 | 6.24 | 3.95 | 42.35 | 14.02 | 1.56 | 0.67 | 2.46 | −0.22 |
| # 70 | | SS | 30–45 | 28.86 | 5.14 | 2.61 | 63.39 | 12.90 | 2.13 | 1.65 | 4.34 | 1.17 |
| Coeff. var. % | | | | 30.76 | 12.03 | 28.25 | 24.37 | 24.37 | 32.87 | 78.94 | 55.26 | 187.13 |

The carbon content in HAs molecules varies rather widely up to 30% and ranges from 14 to 47% C. The HAs samples under study were extracted from the watt area (#22, 24, 25, 27), the zonal tundra (#47, 48), detritus deposit #47 from the watt surface, silty marine deposit #48 from the shallow water and marsh soil (#67, 68, 70). The highest carbon content is noted in the HAs sample of plant detritus, accumulating up to 50% C, which is comparable to the carbon content from natural soils [14]. The lowest content is noted in the HAs of the gley horizon of the Arctic tundra (#46); here, it accumulates up to 13% C.

Relatively low values of the HAs carbon content are characteristic of the waterlogged areas of the Arctic and Subarctic, whose soils develop near the occurrence of permafrost and, in general, under conditions of below-zero temperatures [14,29–31]. It may also be due to the fact that the initial gley horizon also has a relatively low content of bulk organic carbon. Under such conditions, the process of macromolecule condensation is complicated, and the process of hydrogenation in humic acid molecules is intensified [32,33].

The content of hydrogen, nitrogen and oxygen in soils varies within a wide range; this is due to a rather large number of factors that influence the accumulation of these elements in the structure of HAs (hydromorphism, climate, quality of humification precursors and composition of soil microbiota) [30]. The nitrogen content of the studied HAs varies within a smaller range, with a coefficient of variation of 28%, and the content varies from 1.33 to 4.55% N. In general, the nitrogen content is directly related to plant composition (humification precursors) and the rate of the decomposition of the plant sediment at low rates, mainly accumulated aliphatic HA chains [14,29,34]. The highest nitrogen content was noted in samples #67 and #27. The marsh sample (#67) is a poorly compacted peat, consisting of herbaceous residues. Herbaceous vegetation can accumulate up to 12% of proteins and nitrogen-containing compounds, which can accumulate as part of HAs during their transformation (mineralization and humification). Sample #27 is represented by the underlying permafrost ⊥C#~g (suprapermafrost alluvial coarse-humus sediments) with features of gleyization, apparently part of the organic matter during the cryogenic mass transfer that accumulates at the boundary with the permafrost. The change in the hydrogen content of HAs is indirectly related to the quality of humification precursors; moss communities contain more hydrogen than, for example, herbaceous vegetation [13]. The

high oxygen content is associated with the better solubility of oxygen-enriched hydrophilic HAs molecules [35].

To provide a graphical estimation of the hydrogenation/dehydrogenation and oxidation/reduction processes of HAs, a van Krevelen diagram was used (Figure 3).

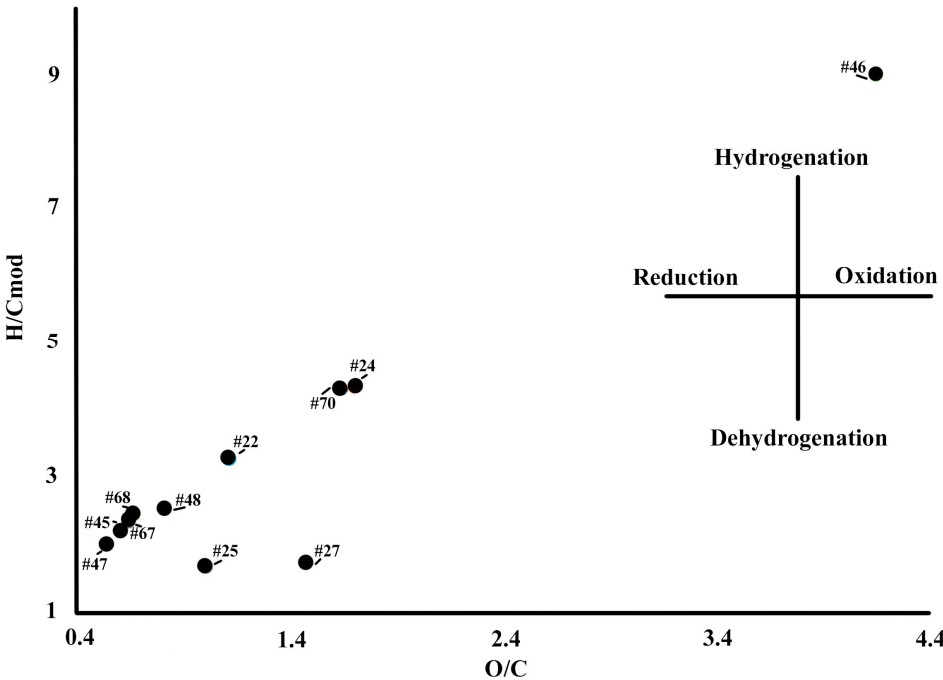

**Figure 3.** Elemental composition of the studied HAs. Soil ID correspond to Table 2. Hydrogenation/Dehydrogenation and Reduction/Oxidation are the processes going in the HAs molecules.

The H/C mod. ratio is an indicator of the stability, or the "maturity", of HAs in soils. The H/C mod. ratio is a more accurate indicator of the degree of stability than H/C because it takes into account the number of substituted hydrogen atoms in the HA structure [31,36]. The lower this value, the less intense the process of attachment of hydrogen to carbon and the development of an aliphatic structure. The data obtained indicate that sample #46 is characterized by the highest H/C ratio, which is apparently associated with a long time of the overmoistening of the Cg horizon in the Arctic tundra conditions and the close occurrence of the permafrost [27,30,35]. The lowest H/C mod. ratio is observed in sample #47 (detritus); this may be due to the quality of the plant material and its diversity.

From the data obtained from the calculation of the W index, it can be determined that most of the studied samples are in oxidizing conditions. Oxidizing conditions indicate active processes of the humification of plant residues in the soil. The weak expression of reduction conditions in the samples (#45, 47, 67, 68) is an indicator of the formation of fresh organic residues and the process of humification in specific bioclimatic conditions [37].

In general, a quite wide variation in the elemental composition data is associated with non-homogeneous biogeochemical conditions in the study sites and the different intensity of biological processes in the studied HAs.

### 3.3. $^{13}$C-NMR Spectroscopy of Humic Acids

According to the data obtained, we can distinguish three main groups of fragments that accumulate in the studied arctic soils, marshes, vats, detritus, and silt; these are C, H-Alkyl ($CH_2$n/CH/C and $CH_3$), aromatic compounds (C-C/C-H, C-O) and OCH groups (OCH/OCq). The obtained spectra are presented in Figure 4. The aromatic group was calculated from the sum of shifts of 110–185 ppm. Aliphatic fragments were calculated from the sum of shifts of 0–110 and 180–200 ppm and AL h, r + AR h, r (total number of

unoxidized carbon atoms). The signals were summed over the regions 0–46 and 110–160 ppm. The chemical shift data are presented in Table 3.

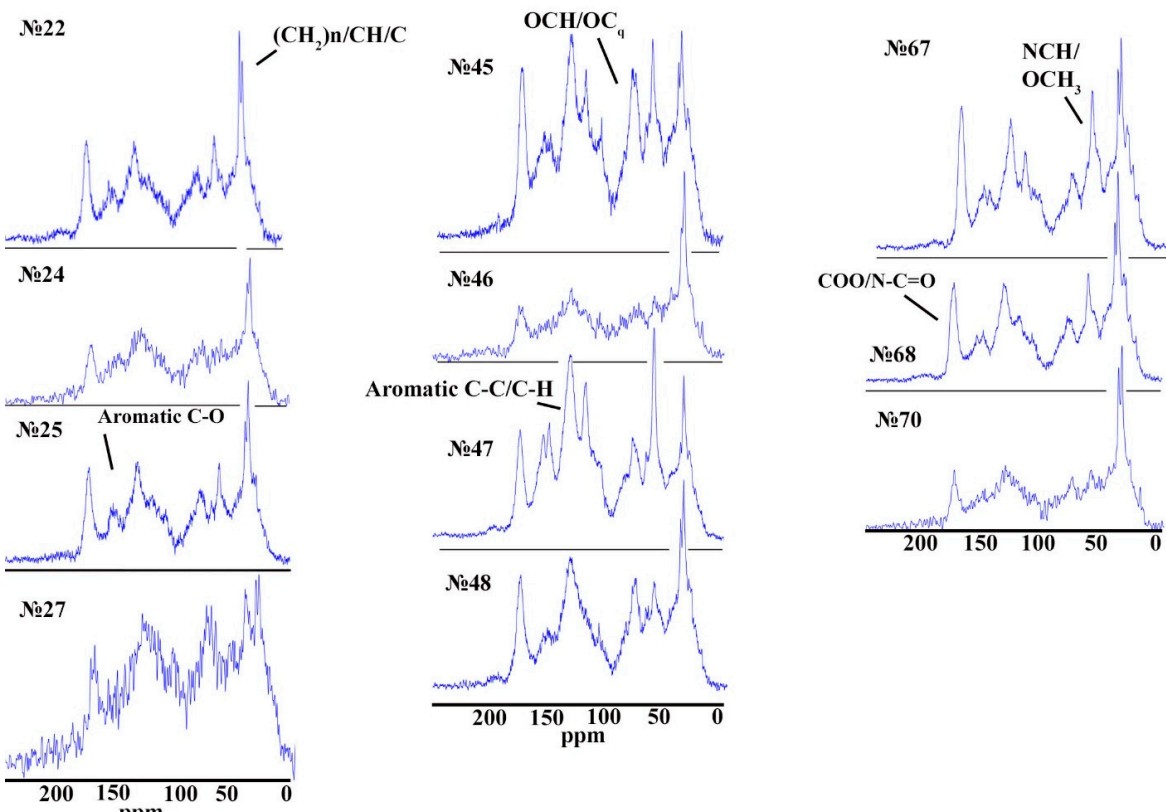

**Figure 4.** $^{13}$C-NMR spectra of the studied HAs. Soil ID correspond to Table 3. The functional groups are presented on subfigure.

**Table 3.** Percentage ratio of the main structural fragments of HAs.

| n/n | Soil Pit Index | Soil Horizon | Depth, cm | Chemical Shifts, % | | | | | | AR | AL | Ar/Al | Al h, r + Ar h, r * | C,H-Al/O, N-Al ** |
|---|---|---|---|---|---|---|---|---|---|---|---|---|---|---|
| | | | | 0–46 | 46–60 | 60–110 | 110–160 | 160–185 | 185–200 | | | | | |
| # 22 | | TE | 0–9 | 34 | 9 | 13 | 26 | 14 | 4 | 40 | 60 | 0.67 | 34 | 9 |
| # 24 | ChGS-19-06 | G | 9–15 | 35 | 8 | 17 | 26 | 11 | 3 | 37 | 63 | 0.59 | 35 | 8 |
| # 25 | | SS | 15–25 | 33 | 9 | 16 | 27 | 12 | 3 | 39 | 61 | 0.64 | 33 | 9 |
| # 27 | | C | 55–75 | 43 | 8 | 13 | 18 | 12 | 6 | 30 | 70 | 0.43 | 43 | 8 |
| # 45 | ChGS-19-14 | TE | 4–8 | 24 | 10 | 24 | 30 | 10 | 2 | 40 | 60 | 0.67 | 24 | 10 |
| # 46 | | G | 8–32 | 44 | 8 | 10 | 22 | 13 | 4 | 35 | 66 | 0.53 | 44 | 8 |
| # 47 | ChGS-19-15 | D (detritus) | 0–5 | 20 | 9 | 23 | 35 | 10 | 3 | 45 | 55 | 0.82 | 20 | 9 |
| # 48 | | H (silt deposit) | 0–5 | 35 | 9 | 13 | 27 | 13 | 3 | 40 | 60 | 0.67 | 35 | 9 |
| # 67 | | TE | 0–5 | 27 | 10 | 21 | 28 | 11 | 3 | 39 | 61 | 0.64 | 27 | 10 |
| # 68 | ChGS-19-23 | G | 5–20 | 35 | 10 | 12 | 26 | 13 | 4 | 39 | 61 | 0.64 | 35 | 10 |
| # 70 | | SS | 30–45 | 41 | 9 | 12 | 23 | 11 | 4 | 34 | 66 | 0.52 | 41 | 9 |

* AL h, r + AR h, r %—degree of hydrophobicity; ** C, H-AL/O,N-AL—degree of organic matter transformation.

Aliphatic structural fragments (55–70%) prevail in the studied HAs samples. The predominance of aliphatic structural fragments in HAs is associated with the low content of aromatic compounds in the composition of humification precursors, as well as with the

low "maturity" of humic substances formed under Arctic tundra, marshes and watt soils. In general, such dynamics are characteristic of excessively humid regions of the Arctic, the wide distribution of reduction conditions and the nearby coast. The input of humic acids from aquatic ecosystems leads to the formation of aliphatic side chains (C-C) and, consequently, to their rapid biodegradation [38–40].

In the conditions of HAs formation, easily hydrolyzable residues are formed from moss residues, algae and various bacteria consisting of proteins, lipids and carbohydrates, which may be subject to relatively rapid biodegradation under the activity of soil microbiota [41]. The studied Cryosols are characterized by a significant variability of microbial biomass content through the profile, but the general regularity of its distribution has maximums in the uppermost horizons, and the sharp decrease in the mineral part of the soil is preserved. It was shown that the maximal fraction of microbial biomass in a mineralizable pool of soil organic matter characterized organomineral horizons, and the minimal one characterized the central and lowermost mineral horizons. The statistical analysis has shown that the contents of soil organic carbon and nitrogen, as well as the soil porosity, had maximal effects on the volume of the microbial biomass and its respiratory activity in organic and mineral horizons of Cryosols [42]. Nevertheless, relative to continental tundra, the coastal region is characterized by a rather high content of aromatic structural fragments; the maximum was established in samples #47 (45%), #45 (40%) and #48 (40%). Samples #47 and # 48 represent sediments of detritus and silt on the coastal surface. The relatively high content of aromatic compounds is apparently related to the composition of the precursors of humification (these sediments consist of herbaceous plant residues) and may be enriched in lignin, which is involved in the formation of aromatic HA fragments during transformation. The content of lignin in deciduous plants ranges from 1 to 22%; in cereal grasses, it ranges from 6 to 10%; in legumes, it is about 6% of dry weight [16]. As for sample # 45, which is associated with the Arctic tundra section, here, the peat includes herbaceous residues, which may be a source of lignin and, correspondingly, aromatic compounds in the HAs. These ratios of aromatic compounds to aliphatic compounds are closest to the soils of the boreal zone [29,43]. Probably, the formation of HAs in littoral soils (in the coastal zone) on watt soils leads to the accumulation of relatively stable organic compounds. In the allochthonous material, humification processes apparently occur during transportation, which cause an increase in the fraction of aromatic compounds. In comparison with the soils of the Lena River delta, we can say that the accumulation of aromatic fragments occurs in soils developing according to a synlithogenic model, relative to zonal cryogenic soils [13].

The other samples studied are characterized by a low degree of accumulation of aromatic compounds in the soil. Apparently, the formation of local marsh and watt soils under reduction conditions and a high degree of hydromorphism leads to the accumulation of C,H-alkyl ($(CH_2)n/CH/C$) chains as well as oxygen-containing fragments (OCH).

Factor analysis was used to analyze the influence of various soil parameters on HAs formation (Figure 5).

Based on the data obtained, we can observe the relationship between the physical parameters of the soil, the pH and the carboxyl, alkyl and quinone groups. The presented cations and anions are combined in one general group and have no connection with other studied parameters, except for $HCO_3$, which is in the C,H-AR, OCH and O,C-Alkyl groups. The formation of HAs is a very complex process in terms of chemistry, so we cannot claim with complete certainty that the studied physico-chemical parameters are decisive in the formation of HAs.

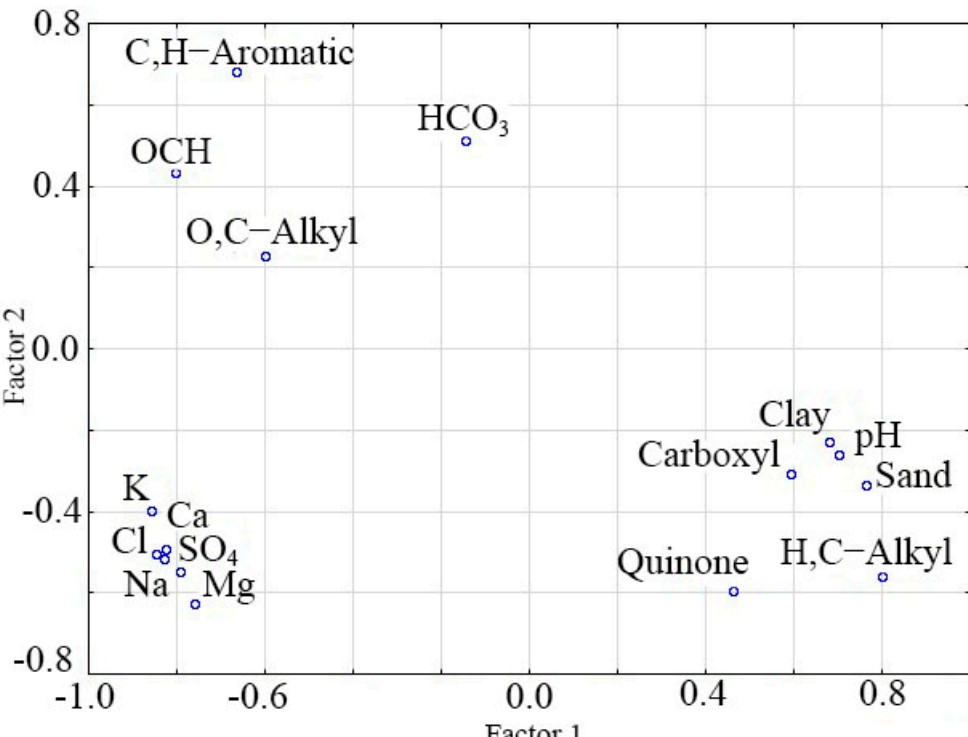

**Figure 5.** Factor analysis among the main soil parameters and functional groups of humic acids. Data of the main soil parameters were published earlier in the article by Gubin et al. [10].

Up to 45% of aromatic compounds were accumulated in the studied samples of HAs, which indicates the stabilization of organic matter in the studied soil and marine sediment samples, but the aliphatic fragments remain dominant. The decrease in the fraction of aromatic fragments is primarily related to reduction conditions as well as to the quality of humification precursors [44,45]. This AR/AL ratio in the samples studied leads to the stabilization and deposition of organic matter in soil and permafrost. Among the samples studied, the greatest contribution to the formation of aromatic structural fragments is made by allochthonous marine material consisting of various kinds of herbaceous plant residues, as well as by herbaceous vegetation forming in the seacoast zone. According to the hypothesis that the wet season promotes the formation of soluble precursors and the dry season promotes molecular condensation, we can assume that, under the conditions of a high degree of hydromorphism, most of the time, soil and soil organic matter are in wet and overmoistened conditions, thereby forming conditions under which the hydrogenation of HAs molecules occurs [14].

The following parameters were used to standardize the quantitative characteristics of HA molecules: the carbon ratio of aromatic structures to aliphatic structures, the degree of organic matter decomposition (C-alkyl/O-alkyl) and the integral hydrophobicity index of HAs (AL h, r + AR h, r) (Figure 6).

According to the data presented in the diagram, it can be noted that the most humified organic acids with a high degree of hydrophobicity accumulate in marsh soils, and the increase in the number of stable HAs molecules is also noted with depth. The increased degree of the hydrophobicity of HA molecules indicates the low availability of this organic matter to microbial decomposition; thus, we can conclude that stable ("mature") organic matter accumulates in marsh soils at the boundary with the permafrost.

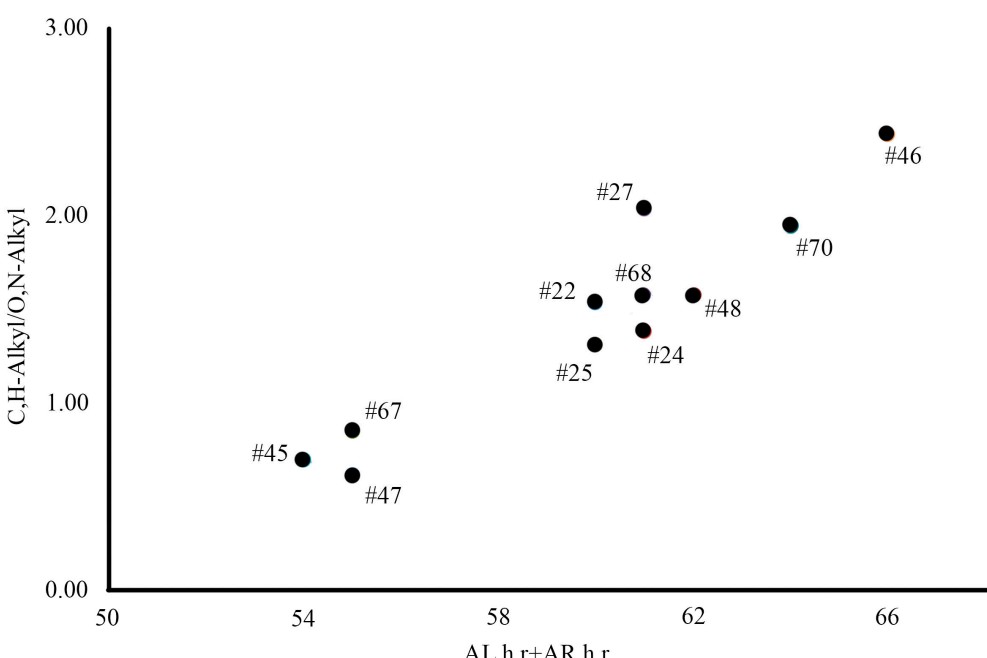

**Figure 6.** Diagram of the integral indicators of the molecular composition of HAs. Soil ID correspond to Table 3.

## 4. Conclusions

The permafrost-affected Cryosols of the accumulative synlithogenic seashores, tidal marshes and watts of the Kolyma Lowland (North Yakutia, Russia) that can extend tens of kilometers from the seacoast were studied. Autochthonous and allochthonous soil organic matter play a leading role in marsh soil formation, including the possible accumulation and biochemical transformation of incoming pollutants (e.g., hydrocarbons). The main objective of this study was to evaluate the soil organic matter genesis and alteration under the influence of tidal processes in coastal permafrost-affected soils, as well as to obtain the previously unknown characteristics of the structural and elemental composition of different fractions of organic matter. It has been established that the highest content of carbon in the composition of HAs is contained in the plant detritus brought with seawater, which forms in the coastal part of the sea and then enters the coastal soils. In the HAs material of soil profiles, the carbon content varies in the range of 30%, which indicates a wide range of conditions of the formation of the organogenic material of the soils that are widespread here. HAs extracted from soils are characterized by a relatively high content of aromatic structural fragments, but aliphatic ones dominate here, which is apparently associated with a high level of soil hydromorphism and the development of reduction conditions in the profiles. The formation of HAs in marsh soils and in watt soils leads to the accumulation of relatively stable organic compounds. In detritus, which represents fresh allochthonous organic material, the humification processes apparently occur during the transport and accumulation of fresh organic matter, which caused an increase in the proportion of aromatic compounds in their structure.

**Author Contributions:** E.A., A.L. and S.G.: conceptualization; E.A.: funding and writing; A.L. and S.G.: expedition with the fieldwork and soil sampling; V.P., E.A. and A.L.: writing of the paper; V.P.: analysis of HAs. All authors have read and agreed to the published version of the manuscript.

**Funding:** This work is supported by the Russian Foundation for Basic Research, project No. 19-05-50107 as well as 21-55-75004 PRISMARCTYC

**Institutional Review Board Statement:** Not applicable.

**Informed Consent Statement:** Not applicable.

**Data Availability Statement:** The data that support the findings of this study are available upon request from the authors and in the Center of Chemical Analyses and Materials and the Center of Magnetic Resonance Research, Scientific Park of Saint-Petersburg State University. The data of the NMR spectroscopy are available from the Center of Chemical Analyses and Materials and the Center of Magnetic Resonance Research.

**Acknowledgments:** The article is devoted to the 300th anniversary of the founding of St. Petersburg State University.

**Conflicts of Interest:** The authors declare no conflict of interest.

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
