# Peer review of "Soil Organic Matter of Tidal Marsh Permafrost-Affected Soils of Kolyma Lowland"

_agronomy, doi:10.3390/agronomy13010048_

Round 1
Reviewer 1 Report
The manuscript reports an accurate and complete analysis of a soil of the Artic sea coast. In the manuscript the authors analyze the soil organic matter of four different sampling points in order to identify its genesis, humification and the influences of tidal processes. Too bad that a microbiological analysis was not also conducted, it would have been very useful for understanding the evolution of organic matter, in particular in anaerobiosis.
I suggest you better describe why you have chosen these four sampling points and define them better accordingly. It may also be useful to enter the coordinates of the single points.
The work is well written. The methodology with which the analyzes were carried out is correct and the processing is well developed.
Author Response
Response to a review of the manuscript “Soil organic matter of tidal marsh permafrost-affected soils of Kolyma lowland”.
Dear reviewer!
Thank you for your comments, they were completely taken into account, which improved the quality of the article for publication in Journal.
Text that has been changed is marked by yellow color.
General comments:
- The manuscript reports an accurate and complete analysis of a soil of the Artic sea coast. In the manuscript the authors analyze the soil organic matter of four different sampling points in order to identify its genesis, humification and the influences of tidal processes. Too bad that a microbiological analysis was not also conducted, it would have been very useful for understanding the evolution of organic matter, in particular in anaerobiosis.
I suggest you better describe why you have chosen these four sampling points and define them better accordingly. It may also be useful to enter the coordinates of the single points.
The work is well written. The methodology with which the analyzes were carried out is correct and the processing is well developed.
Response: Thank you! The microbiological analysis was done earlier and the results have already been published, we have expanded section 3.3 and added information about the microbiological analysis with a link to the relevant work. We have improved the Study area section and added more information about selecting sampling locations, as well as sampling coordinates.
Thank you for work of our article.
Sincerely,
Engineer of Saint-Petersburg State University, Vyacheslav I. Polyakov.
Professor of Saint-Petersburg State University, Evgeny V. Abakumov.
Reviewer 2 Report
The following are my suggestions:
Materials and Methods:
Please state the reasons for choosing these soil pits? How to prove the representativity of the soil samples? Any duplicates?
R&D:
Please explain the relationship between humification and aromaticity of SOM. How does humification process affect aromaticity? (L444-447) How much does the composition of the precursors of humification contribute to aromaticity? (L376-378)
Please clarify the full name of the abbreviates when they first appear, such as TDS.
Please check the format of the chemical formula, for instance, (CH2)n/CH/C should be (CH2)n/CH/C .
Author Response
Response to a review of the manuscript “Soil organic matter of tidal marsh permafrost-affected soils of Kolyma lowland”.
Dear reviewer!
Thank you for your comments, they were completely taken into account, which improved the quality of the article for publication in Journal.
Text that has been changed is marked by yellow color.
General comments:
Please state the reasons for choosing these soil pits? How to prove the representativity of the soil samples? Any duplicates?
R&D:
Please explain the relationship between humification and aromaticity of SOM. How does humification process affect aromaticity? (L444-447) How much does the composition of the precursors of humification contribute to aromaticity? (L376-378)
Please clarify the full name of the abbreviates when they first appear, such as TDS.
Please check the format of the chemical formula, for instance, (CH2)n/CH/C should be (CH2)n/CH/C.
Response: We have improved the Study area section and added more information about selecting sampling locations, as well as sampling coordinates. The elemental composition was done in three replicates. NMR spectroscopy method is based on obtaining a large number of scans from which later NMR spectra are obtained.
The conclusions were reworked; we tried to avoid a direct connection between humification and the increase in aromatic structural fragments. This is due to different hypotheses of humification, according to the polymeric model, we could say that under conditions of active humification and condensation of macromolecules the formation of aromatic structural fragments occurs. However, from the point of view of the supramolecular theory, under conditions of humification and transformation of organic matter, there is an increase in low-molecular-weight compounds in the composition of humic acids.
We have added additional information about the lignin content of various humification precursors.
We have checked the abbreviations and chemical formulas.
Thank you for work of our article.
Sincerely,
Engineer of Saint-Petersburg State University, Vyacheslav I. Polyakov.
Professor of Saint-Petersburg State University, Evgeny V. Abakumov.
Reviewer 3 Report
Dear Authors!
Thanks for giving me the opportunity for evaluation of the manuscript entitled " Soil organic matter of tidal marsh permafrost-affected soils of Kolyma lowland".
The manuscript has been well written, the story has been well portrayed in the draft. However, it needs to be check for some minor English grammar and spelling mistakes.
Author Response
Response to a review of the manuscript “Soil organic matter of tidal marsh permafrost-affected soils of Kolyma lowland”.
Dear reviewer!
Thank you for your comments, they were completely taken into account, which improved the quality of the article for publication in Journal.
Text that has been changed is marked by yellow color.
General comments:
Thanks for giving me the opportunity for evaluation of the manuscript entitled " Soil organic matter of tidal marsh permafrost-affected soils of Kolyma lowland".
The manuscript has been well written, the story has been well portrayed in the draft. However, it needs to be check for some minor English grammar and spelling mistakes.
Response: Thank you! We checked English grammar and spelling mistakes.
Thank you for work of our article.
Sincerely,
Engineer of Saint-Petersburg State University, Vyacheslav I. Polyakov.
Professor of Saint-Petersburg State University, Evgeny V. Abakumov.
Author Response
Response to a review of the manuscript “Soil organic matter of tidal marsh permafrost-affected soils of Kolyma lowland”.
Dear reviewer!
Thank you for your comments, they were completely taken into account, which improved the quality of the article for publication in Journal.
Text that has been changed is marked by yellow color.
General comments:
Line 18 Kindly replace ‘plays here’ with “play”.
Response: It was replaced.
Line 64 Kindly replace ‘migrates’ and ‘accumulates’ with “migrate” and
“accumulate.
Response: It was replaced.
Line 135 m.a.s.l must be written in full
Response: It was rewritten
Line 179 – 180 Kindly re-write the sentence to make it meaningful.
Response: The sentence has been rewritten
Line 187 Kindly define what HA means
Response: In section MM we have deciphered the term.
Line 337 there is the need to explain or indicate what the ‘Windex’ is. This was
not indicated in the materials and methods or mentioned as part of the
indices to be used in any assessment.
Response: We indicate the term in MM section.
Line 383 Kindly replace ‘probability’ with “probably”
Response: It was replaced.
Line 427 Kindly replace ‘plays here’ with “play”
Response: It was replaced.
Line 432 – 435 I suggest that this section be moved to the materials and methods
section.
Response: The information has been deleted
The data availability statement needs to be re-written. The statement in its current form is not very clear.
Response: The section has been rewritten.
Thank you for work of our article.
Sincerely,
Engineer of Saint-Petersburg State University, Vyacheslav I. Polyakov.
Professor of Saint-Petersburg State University, Evgeny V. Abakumov.
Reviewer 5 Report
Soils of the Arctic sea coasts are diverse and unique. It is an important work to get the attention of more scholars/research on soils in this area. In this study, SOC genesis information along soil profile are presented by HA compositions and by 13C-NMR methods. The morphology of studied area is also detailed described. Besides analysis listed in this manuscript, factor analysis of SOM compositions with topography, morphology, soil basic properties (such as soil pH, EC, soil temperature, etc.) and soil layer depth along profile should be conducted to reveal the possible mechanism of SOM formation in studied area. Thus, I recommend to add soil basic properties firstly. Then further statistics should be conducted in revised manuscript.
Author Response
Response to a review of the manuscript “Soil organic matter of tidal marsh permafrost-affected soils of Kolyma lowland”.
Dear reviewer!
Thank you for your comments, they were completely taken into account, which improved the quality of the article for publication in Journal.
Text that has been changed is marked by yellow color.
General comments:
Soils of the Arctic sea coasts are diverse and unique. It is an important work to get the attention of more scholars/research on soils in this area. In this study, SOC genesis information along soil profile are presented by HA compositions and by 13C-NMR methods. The morphology of studied area is also detailed described. Besides analysis listed in this manuscript, factor analysis of SOM compositions with topography, morphology, soil basic properties (such as soil pH, EC, soil temperature, etc.) and soil layer depth along profile should be conducted to reveal the possible mechanism of SOM formation in studied area. Thus, I recommend to add soil basic properties firstly. Then further statistics should be conducted in revised manuscript.
Response: Thank you! The basic soil parameters have already been published before, we refer to this work in our article. We performed a factor analysis between the main soil parameters and the content of structural fragments in GC. The information is presented in section 3.3.
Thank you for work of our article.
Sincerely,
Engineer of Saint-Petersburg State University, Vyacheslav I. Polyakov.
Professor of Saint-Petersburg State University, Evgeny V. Abakumov.